# Tunable Terahertz Wavefront Modulation Based on Phase Change Materials Embedded in Metasurface

**DOI:** 10.3390/nano12203592

**Published:** 2022-10-13

**Authors:** Ming Zhang, Peng Dong, Yu Wang, Baozhu Wang, Lin Yang, Ruihong Wu, Weimin Hou, Junyao Zhang

**Affiliations:** School of Information Science and Engineering, Hebei University of Science and Technology, Shijiazhuang 050018, China

**Keywords:** metasurfaces, phase change materials, terahertz (THz) region, PB phase, tunable wavefront manipulation

## Abstract

In the past decades, metasurfaces have shown their extraordinary abilities on manipulating the wavefront of electromagnetic wave. Based on the ability, various kinds of metasurfaces are designed to realize new functional metadevices based on wavefront manipulations, such as anomalous beam steering, focus metalens, vortex beams generator, and holographic imaging. However, most of the previously proposed designs based on metasurfaces are fixed once design, which is limited for applications where light modulation needs to be tunable. In this paper, we proposed a design for THz tunable wavefront manipulation achieved by the combination of plasmonic metasurface and phase change materials (PCMs) in THz region. Here, we designed a metal-insulator-metal (MIM) metasurface with the typical C-shape split ring resonator (CSRR), whose polarization conversion efficiency is nearly 90% for circular polarized light (CPL) in the range of 0.95~1.15 THz when PCM is in the amorphous state, but the conversion efficiency turns to less than 10% in the same frequency range when PCM switches into the crystalline state. Then, benefiting from the high polarization conversion contrast of unit cell, we can achieve tunable wavefront manipulation by utilizing the Pancharatnam–Berry (PB) phase between the amorphous and crystalline states. As a proof-of-concept, the reflective tunable anomalous beam deflector and focusing metalens are designed and characterized, and the results further verify their capability for tunable wavefront manipulation in THz range. It is believed that the design in our work may pave the way toward the tunable wavefront manipulation of THz waves and is potential for dynamic tunable THz devices.

## 1. Introduction

Metasurfaces, artificially designed two-dimensional metamaterials [1], can flexibly manipulate the wavefront of beam to change the propagation of beam [2]. Therefore, the metasurfaces have been widely investigated and developed in wavefront modulation devices, such as anomalous beam deflection [3], light focusing [4], vortex beams generator [5], holographic imaging [6,7,8], and polarization conversion [9]. Compared with the three-dimensional metamaterials, the metasurfaces are composed of planar meta-atoms with specific electromagnetic response in a certain order, which has less thickness, lower ohmic losses, simpler design and manufacturing process. Through the interaction between electromagnetic waves and meta-atoms, metasurfaces can fully utilize and manipulate the abrupt phase changes of beam to control wavefront [10]. These characteristics grant metasurfaces many great potential applications. With continuous in-depth research of terahertz technology, various metasurface devices that manipulate THz wavefronts have been increasingly proposed [11,12,13]. However, the functions and capabilities of the most metasurfaces are fixed when their design is complete, and thus, they are difficult to apply in fickle situations and not suitable for applications that needs tunable wavefront manipulation [14].

In recent years, metasurfaces combined with phase change materials (PCMs) to achieve dynamic photonic devices have become a research hotspot [15,16]. Phase change material (PCM) is a promising and earth-abundant alternative to the next-generation nonvolatile optical devices, offering a new avenue to realize tunable wavefront manipulation. GeSbTe (GST) alloys as typical phase change materials have been used for many years in optical disk storage [17,18] and have been introduced to reconfigurable photonic devices recently. Compared to phase change material VO_2_ [19], GST [20] has greater advantages in stability, conversion rate [21], and non-volatility [22]. GST can be converted between two states (usually amorphous and crystalline state) or the intermediate states by external excitation (such as thermal [23], electrical [24], and optical excitation [25]). The transformation of PCMs between the amorphous and crystalline states will bring significant difference in optical and electrical properties. In addition, the GST exhibits high refractive index and contrast and lower absorption loss in THz region [21,26,27], which enables it to be integrated into THz devices to achieve tunable functions. GST alloys are ideal materials for switchable or reconfigurable devices, such as thermal emitters [28], Fresnel zone plates [29], and absorbers [30]. Furthermore, several works [31,32,33] have shown great potential of GST in dynamic wavefront manipulation. Among them, PCMs are integrated in THz switchable metalens, optical vortex generators and beam steering verified tunable wavefront manipulation [34,35], and experimental verifications were carried out. However, the designs of terahertz tunable metasurfaces are complex [36] and difficult to extend to other types of photonic devices, which hinders the further development of THz tunable metadevices.

In this work, the phase change material GST was embedded into the plasmonic metasurface to realize tunable wavefront manipulation of THz waves. The design in our work is technically challenging as it requires high contrast of polarization conversion between two states. We utilized the genetic algorithm (GA) to optimize the geometric parameters of the unit cell and to obtain high contrast response in two states. The design was a metal-insulator-metal (MIM) configuration with the typical C-shape split ring resonator (CSRR). The unit cell shows high circular polarization conversion efficiency near 90% at amorphous state but low conversion efficiency less than 10% at crystalline state at a frequency of 0.95~1.15 THz. To verify the capability of the proposed design for tunable wavefront manipulation, we designed tunable anomalous beam deflector and focusing metalens by rigorously arranging unit cell with PB phase. In amorphous state, these two metadevices can deflect and focus THz beams with high efficiency. However, in crystalline state, the metadevices act as conventional reflective mirrors. Furthermore, both metadevices show broadband characteristics. It is believed that the tunable metasurface will be further developed in the field of THz beam manipulation.

## 2. Design and Methods

The split-ring resonator is a kind of magnetic metamaterial. In 1981, Hardy described the magnetic split-ring resonance of a hollow cylinder with a linear notch at about 1 GHz [37]. Metal ring is equivalent to the inductance to produce an induced electromagnetic field in a variable magnetic field perpendicular to it, and the gap of split-ring regarded as capacitance is introduced to produce resonance. Therefore, the incident changing electric field forms magnetic field polarization on the metal ring, which causes induced circulation on the metal ring, and the charge accumulates at a gap between both ends, so that the electric field energy is accumulated at the ring. Now, this design is used as a prototype of many metadevice units in metasurface research.

The schematic diagram of the metasurface for tunable wavefront manipulation is shown in Figure 1. The unit cell is a metal-insulator-metal (MIM) configuration with CSRR. The materials of the top-layer CSRR array and bottom metal reflective ground are pure copper, and the intermediate dielectric layer is phase change material GST. In addition, the bottom metal reflection plate plays a role in reflecting the transmitted light, and the GST dielectric layer plays a role in regulating the dielectric constant to change the electromagnetic response of the metasurface. According to the different lattice state (crystalline and amorphous state) of the GST, metasurface can exhibit different electromagnetic response (cross-polarization or co-polarization reflection) for the circular polarization light (CPL).

For such an anisotropic unit cell, if left-handed or right-handed circular polarization light $E_{I}^{R/L}$ is incident to metasurface from −*z* direction, the scattering light $E_{S}^{R/L}$ can be described as [38].
(1)$$E_{S}^{R/L}=\frac{t_{o}+t_{e}}{2}E_{I}^{R/L}+\frac{t_{o}-t_{e}}{2}\exp(\mp i2\varphi )E_{I}^{L/R}$$

The first term in Formula (1) represents the transmitted light having the same handedness as the incident light, and the second term represents the scattering light having the opposite handedness with respect to the incident light and attaching a Pancharatnam- Berry (PB) phase of ∓2*φ* (“−“ for right-handed and “+” for left-handed circularly polarized incident light), where *t*_o_ and *t*_e_ are the coefficients of incident linearly polarized light along the long axis and short axis of anisotropic structure, respectively [2]. Benefitting from the significant differences of dielectric constant of GST in two states, we can control the *t*_o_ and *t*_e_ to make the scattering light only have the cross-polarized or co-polarized part by adjusting the geometric parameters of unit cell. As illustrated in Figure 1, when GST is in amorphous state, the incident left-handed or right-handed circular polarization light is converted to its cross polarization by metasurface. Meanwhile, scattering light will produce an additional abrupt phase, and the value of the phase is a function of the rotation angle *φ* of the CSRR. When GST is in crystalline state, the handedness of scattering light is the same as incident circular polarized light. When the CSRR rotates from 0 to π, the phase can be tuned from 0 to 2π for the opposite handedness. Moreover, the geometrical shape of each unit cell remains essentially unchanged, only changing the rotate angle *φ* with respect to the *x*-axis, so the transmission amplitude remains almost unchanged. Based on this characteristic, phase is adjusted according to different rotation angles of the CSRR, so as to manipulate the wavefront of THz waves. Meanwhile, by adjusting the lattice state of GST, one designed metasurface can achieve two functions.

The real and imaginary part of permittivity of GST in amorphous state and crystalline state are shown in Figure 2a. The permittivity is fitted to the range of 0.1~2 THz from the measured data of visible light and infrared by CST 2018 software (CST, Germany) utilizing enough measurements, and the result is consistent with the previous papers [39,40]. Meanwhile, the measurement values were measured by spectroscopic ellipsometer (SENTECH SE850 and SENDIRA) in visible and infrared bands [41]. In the wide frequency region, we can see that the real part of the permittivity of GST shows significant difference between amorphous and crystalline states. The imaginary part of the permittivity in both states is close to 0, reflecting excellent feature of low absorption loss of GST, which is vital for metadevices with high efficiency. The material of pure copper was loaded from the library, whose electric conductivity is 5.96 × 10^7^ s/m. Periodic boundary conditions were applied in the *x* and *y* directions while perfectly matched layer boundary condition was used in the *z* direction.

Numerical simulations are performed using the finite element method (FEM) in a commercial software package CST Microwave Studio 2018. Meanwhile, we used genetic algorithm to optimize the geometric parameters of the unit cell and to obtain the flat response with frequency and 2π phase shift. By optimizing the geometric parameters, the unit cell can realize different polarization conversion between amorphous and crystalline states. The main operators of genetic algorithm are selection, crossover, and mutation and genetic algorithm is also widely used in optimization, planning, design, etc. Although CST Microwave Studio (CST) software has its own genetic algorithm optimization function, it is found that its function is relatively fixed and lacks flexibility through our attempts, and it is not convenient to process the simulation results data. Here we used the co-simulation of MATLAB R2016b software (MathWorks, USA) and CST 2018 software, which means MATLAB and CST can establish communication links. Then the genetic algorithm code program is written by MATLAB. The process of executing the program is to control CST modeling and simulation, and the simulation results data can also be accessed by MATLAB, which is automatic and efficient (more detailed setup of simulation is presented in Appendix A).

After the rough model of the metasurface unit cell is determined, a set of optimal parameters needs also to be simulated to determine the final unit structure. In order to avoid a lot of time and energy consumption by manual parameter setting, we use genetic algorithm to automatically optimize a set of optimal parameters, and just need to focus on the final result. Meanwhile, the optimal parameters represent the final unit structure showing high polarization conversion efficiency and wide working band-width. As is illustrated in Figure 2, the unit cell shows high circular polarization conversion efficiency near 90% at amorphous state but low conversion efficiency less than 10% at crystalline state in the frequency of 0.95~1.15 THz. In the initial population, each unit cell is determined by five parameters of [p, h, α, Rin, Rout] (Figure 1). The fitness function is shown as
(2)$$\begin{array}{l} F(X)=[w_{1},w_{2},w_{3},w_{4}]\;*\;\left[\begin{array}{l} \sum_{i=1}^{1001}f(0.9\leq x\leq 1.0)\\ \sum_{i=1}^{1001}f(0.8\leq x<0.9)\\ \sum_{i=1}^{1001}f(0.7\leq x<0.8)\\ \sum_{i=1}^{1001}f(0.6\leq x<0.7) \end{array}\right]\\ X=\left\{p,h,\alpha ,R_{in},R_{out}\right\}\in \left\{\left[45,65],[15,35],[55,75],[8,12],[13,17\right]\right\} \end{array}$$
where *f(x)* is used to judge the polarization conversion efficiency *x* and obtain a value of 0 or 1. The *i* represents a certain point in the dataset, and [*w*_1_, *w*_2_, *w*_3_, *w*_4_] are the weight coefficients. According to the requirement of this work, the weight coefficients are [1,4,7,12].

The genetic algorithm is set as follows: the size of populations is 40 in a generation and the crossover probability is set as 0.8, the mutation rate is set as 0.01. Simultaneously, the fitness value is normalized and recorded in Figure 3a, where the trend of mean fitness is incremental. Figure 3a shows the fitness value of the optimal individual has not changed since the 5th generation, that is, the optimal individual has been obtained in the 5th generation. After 7 generations of iterations, the stability of the optimal results is fully verified. The parameters of optimized individual are shown in Figure 3b and the result is [55 μm, 20 μm, 65°, 10 μm, 14 μm] [42].

Normally incident plane wave with left circular polarization was used to excite the metasurface. The cross-polarized and co-polarized reflectance of the scattered light under the amorphous and crystalline state were recorded, which is shown in Figure 2b,c. In the region of 0.95~1.15 THz, and when GST is in amorphous state, cross-polarization reflectance is nearly 90%, at the same time, co-polarization reflectance is close to 0. On the contrary, when GST is in crystalline state, cross-polarization reflectance is close to 0, and the co-polarization reflectance is more than 90%. Figure 3c shows the relationship of reflectance and phase shift with respect to the orientation angles. R_cross_a_ and PS represent the cross-polarized reflectance and phase shift at 1.1 THz in amorphous state. We can see that when *φ* is varied, the phase shift is very consistent with the PB phase of 2*φ*, and cross-polarized reflectance in amorphous state R_cross_a_ maintains high efficiency. We can also see that the unit cells keep low cross-polarized reflectance in crystalline state at same frequency from the red line of R_cross_c_. The significant difference in spectral response between the two states exhibits the capability of the designed metasurface to achieve tunable wavefront manipulation.

In addition, we calculate the polarization conversion ratio (PCR = R_cross_/(R_cross_ + R_co_)) to characterize the operating bandwidth. Figure 4a shows the simulated PCR of two states, in the region of 0.95~1.15 THz, respectively. PCR is more than 90% in amorphous state, even close to 100% in 0.95~1.1 THz, and less than 10% in crystalline state. So, the operating bandwidth of the tunable metasurface is 0.95~1.15 THz (defined as PCR exceeding 90% in amorphous state and below 10% in crystalline state). To further explore the physical mechanism of different responses of unit cell in GST two states, the reflective phase difference between two LP components along the *x* and *y* directions are calculated as depicted in Figure 4b. For a unit cell illuminated by normally incident linearly polarized light with a polarization angle of 45°, a phase difference with a slight variation in the region of 0.9*π*~1.1*π* exists, which is a requirement to convert circularly polarized light to its cross-polarization state. This result shows the operating bandwidth of 0.97~1.14 THz in amorphous state. However, the phase difference between two orthogonal components in crystalline state cannot meet the condition of polarization conversion resulting in the most deflection of co-polarization.

In order to further reveal the physical mechanism of the unit cell producing opposite electromagnetic responses in GST two states, we illustrate the instantaneous electric field distributions and surface current under normal incidence. Figure 5a,b illustrated the opposite electromagnetic responses of the two devices under different states. In amorphous state, the designed unit cell can realize the circular polarization conversion. In crystalline state, the unit cell works as a mirror-like device. Figure 5c,d shows the distributions of electric field **E**x at a resonant frequency of 1.1 THz in the *xy* plane, respectively. The direction of arrow represents the vibration direction of electric field vector. It is obvious that the electric fields are highly localized in the opening of the split ring in both states. The vibration direction of electric field vector rotates at the opening of the split ring in amorphous state, but in crystalline state, the electric field directions on both sides of the opening of the split ring are opposite. The different electric field distributions lead to diametrically opposite polarization conversion efficiencies in the two states. Figure 5e,f show the distributions of surface current at a resonant frequency of 1.1 THz in the *xy* plane, and the direction of arrow represents the direction of surface current. In amorphous state, the surface currents have the same circulating direction along the split ring, corresponding to the symmetric resonance mode. Since the direction of the surface current in the amorphous state is the same as that on the split-ring resonator, the incident circularly polarized light will be deflected along the direction of the surface current to realize polarization conversion. But in crystalline state, the surface currents exhibit both counterclockwise and clockwise directions along the split ring, corresponding to the asymmetric resonance mode. The asymmetric resonance mode induces the magnetic resonance [43,44]. Therefore, the components of the incident circularly polarized light deflecting along the left and right sides of the split ring resonator will repeal by implication, and THz waves are simply specular reflected. The much larger refractive index of GST suppresses the transversal coupling. Thus, the unit cell exhibits a weak anisotropy and low polarization conversion efficiency. Based on this characteristic, the tunable wavefront manipulation can be realized by appropriately arranging the unit cell.

## 3. Characterization

According to the design and analysis of the above unit cell, wavefront manipulation can be achieved only by arranging the unit cell with specific requirements. Therefore, metasurfaces can be designed for various metadevices. To demonstrate the tunable wavefront manipulation, we designed and simulated two reflective tunable metadevices. The phase gradient induced by a linear change of orientation with the coordinates produces a helicity-dependent transverse wave vector. Thus, the planar anisotropic metasurface can deflect the propagation direction of THz beams under the normal right- and left-hand polarized incidence. The designed metadevice consists of periodic arrays of 15 unit cells with an incremental rotation angle of *π*/15, and the rotation is clockwise along the *x*-axis, as shown in Figure 6a,b. In the simulations, the two-dimensional beam deflector was periodically extended along *x*-axis and *y*-axis by setting the boundary condition as periodic. When GST is in amorphous state, the metadevice exhibits an anomalous beam deflector with design angle *θ_r_*, where the deflection angle *θ_r_* is described by the generalized Snell law [45,46]
(3)$$\sin\theta_{r}=\frac{\lambda \Delta \phi }{2\pi d}+\sin\theta_{i}$$
where *λ* is the operating wavelength, *d* is the distance between unit cells. Δ*ϕ* is the phase gradient between each unit cell, which is determined by the number of abrupt phase sampling points, so *θ_r_* can be manually adjusted by the number of unit cells. Here, the designed anomalous deflection angle *θ_r_* = 19.36°. In Figure 6a,b, we show the reflection intensity of the metadevice at the frequency of 1.1 THz in amorphous and crystalline states, respectively. When GST is in amorphous state, the deflection angle *θ_r_* is 18°, which is close to the theoretical value, and the reflection intensity is more than 80%. When GST is in crystalline state, the beam deflector is equivalent to an ordinary mirror. Thus, according to conventional deflection law, deflection angle *θ_r_* equals to the incident angle *θ_i_* (i.e., *θ_r_* = *θ_i_* = 0°), and the reflection intensity exceeds 95%. Figure 6c,d show the instantaneous normalized electric field distribution under normal incidence light of left-handed circularly polarized (LCP). The results further confirmed the anomalous deflection in amorphous GST and normal specular deflection in crystalline GST of the metadevices, which also exhibits its capability of the tunable wavefront manipulation.

To demonstrate the bandwidth feature, the anomalous beam deflector was simulated under the illumination of LCP light at the frequency ranging from 0.95~1.15 THz. Normalized far-field radiation patterns in two states are presented in Figure 7. From Figure 7a, the incidence lights in the GST amorphous state are deflected anomalously at the angle of 22.57°~18.48° in the whole frequency range. Figure 7b shows that incidence lights in GST crystalline state are deflected in the conventional mirror in the same frequency band. These two figures show that the reflection intensity of amorphous and crystalline state is roughly more than 80% and 90% in the whole frequency range, respectively. The results further indicate the broadband characteristics of anomalous beam deflector designed by this paper and make the metadevice more convenient for application.

To further demonstrate the ability of tunable wavefront manipulation, the unit cells are arranged to construct a reflective tunable focusing metalens. The area of metalens is 1155 × 1155 μm^2^, and the focal length is set as 500 μm. To focus light correctly, the model of focusing metalens is composed of central unit cell clockwise rotation along *x*-axis and *y*-axis, respectively. The phase difference on the metadevice can be expressed as [47]
(4)$$\Phi (x,y)-\Phi (O)=\frac{2\pi }{\lambda }(\sqrt{x^{2}+y^{2}+f^{2}}-f)$$
where *Φ*(*x,y*) represents the phase distribution of the unit cells with respect to center *O*. *λ* is the operating wavelength, and *f* represents design focal length of the focusing metalens. Figure 8a–d show the distribution of electric field intensity on *y* = 0 μm plane and *z* = 500 μm focal plane in both GST states, respectively. When GST is in amorphous state, as shown in Figure 8a,b, 1.05 THz LCP light wavefront becomes spherical after the phase is manipulated and beam is reflected by focusing metalens, so most of the right-handed circularly polarized (RCP) light is focused to a point. When GST is in crystalline state, owing to when the LCP light passes through the focusing metalens, reflected light always has a certain RCP light component. Therefore, as shown in Figure 8c,d, RCP light also has a weak focus when it is reflected by focusing metalens. However, Figure 8e shows the difference of focusing intensity between amorphous and crystalline states of GST is obvious and more than 40 times contrast. This not only verifies the tunable wavefront manipulation ability of focusing metalens, but also verifies capability of the design and realizes beam focusing.

When the frequency of incident circularly polarized light changes, to make focusing metalens cover 0~2π phase, the rotation angle *φ* of unit cells should be changed, to constitute different metalens for different incidence light frequencies. In this paper, we redesign focusing metalenses for LCP light with the frequency of 1 THz, 1.1 THz, and 1.15 THz. As shown in Figure 9, in GST amorphous state, LCP light on these four metalenses has cross-polarization, and the spherical wavefront constituted by additional phase makes the beam to focus to a point. When GST is in crystalline state, four metalenses behave as mirror-like reflector. The results exhibit the tunable and broadband characteristics of the focusing metalens.

## 4. Conclusions

In this paper, the tunable wavefront manipulation of terahertz electromagnetic wave is realized via the combination of plasmonic metasurface with phase change material GST. The optimization of enabling wavefront manipulation in one state and suppressing it in another state is sophisticated and require elaborate design to make a balance between the two states. The design in our work is technically challenging as it requires high contrast of polarization conversion between the two states. We utilized genetic algorithm (GA) to optimize the geometric parameters of the unit cell and to obtain the high contrast response in two states. By embedding the GST into a typical MIM structure, the CSRR shows that when GST changes from crystalline to amorphous at 0.95~1.15 THz, the cross-polarization reflectance changes from less than 10% to nearly 90%, which indicates high polarization conversion contrast and wide operating bandwidth. The same basic structure could be redesigned for other NIR and MIR spectral bands, where GST exhibits low absorption losses and high refractive index contrast. Moreover, on the basis of the proposed method, independent beam manipulation in more than two different states can be further achieved. Then, reflective tunable anomalous beam deflector and focusing metalens based on PB phase are designed to demonstrate the capability of tunable wavefront manipulation. The two metadevices show obvious contrast in EM response in amorphous and crystalline state. Moreover, the wavefront manipulation achieved by the anisotropy structure based on PB phase can be extended to apply for other types of dynamic photonic devices, such as dynamic hologram imaging and dynamic vortex beam generators. The designed metasurfaces-based devices show great advantages in THz region manipulation and solve the problems of complex design and large volume of traditional THz manipulation system. Therefore, it is believed that the design in our work may pave the way toward the tunable wavefront manipulation of THz electromagnetic waves applied in beam-steering and beam-shaping applications.

## Figures and Tables

**Figure 1 nanomaterials-12-03592-f001:**
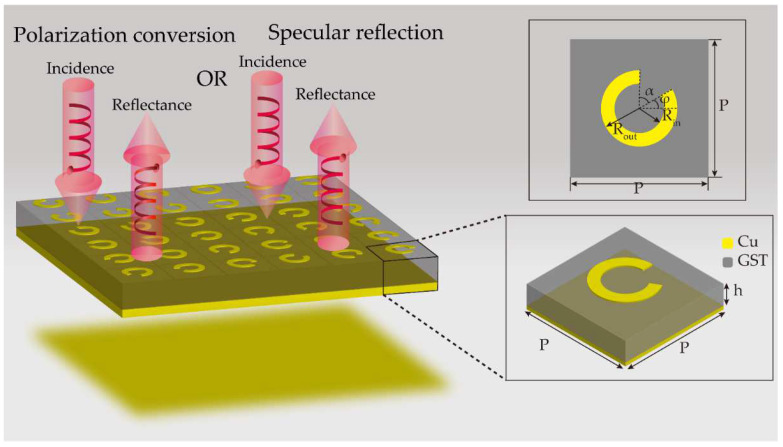
The schematic diagram of the metasurface for tunable wavefront manipulation. The insets are the top view and 3D illustration of unit cell with its geometric parameters. The geometric parameters of the unit cell: period *p*, the thickness of GST *h*, the opening angle *α*, the rotate angle with respect to the *x*-axis *φ* = 0° initially, the inner radius *R_in_*, the outer radius *R_out_*, the thicknesses of the top-layer CSRR and the bottom metal reflector plate are 0.2 μm and 10 μm, respectively.

**Figure 2 nanomaterials-12-03592-f002:**
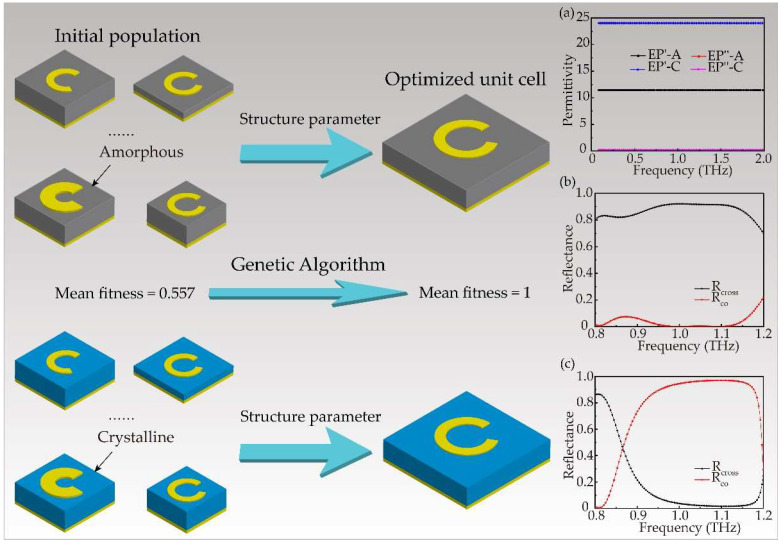
Schematic diagram of genetic algorithm. The genetic algorithm optimizes the CSRR unit structure with high polarization conversion efficiency and wide working band-width. (**a**) The real and imaginary parts of permittivity of GST in two states, A and C represent the amorphous and crystalline states, respectively. The simulated results of cross-polarization and co-polarization reflectance in amorphous state (**b**) and crystalline state (**c**).

**Figure 3 nanomaterials-12-03592-f003:**
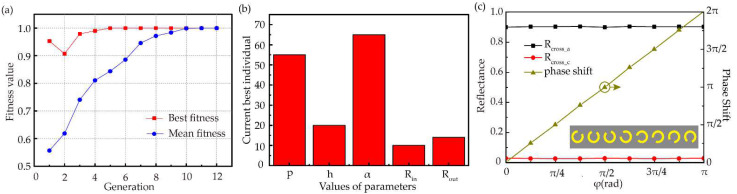
(**a**) The variation of fitness value with generation in the optimization. (**b**) The values of the parameters for the best optimized individual. (**c**) Simulated cross-polarization reflectance and phase shift as a function of rotation angle *φ* in amorphous state and cross-polarization reflectance in crystalline state. The deep yellow line corresponds to the vertical axis on the right, and the unit with different rotation angles in the inset represent different phase shifts.

**Figure 4 nanomaterials-12-03592-f004:**
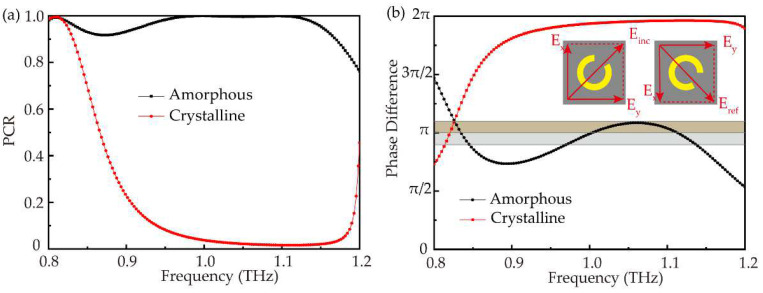
(**a**) The PCR with respect to the frequency in two states. (**b**) The calculated phase difference between the two orthogonal polarizations in two states (the dark yellow and grey band represent the slight variation of phase difference of π).

**Figure 5 nanomaterials-12-03592-f005:**
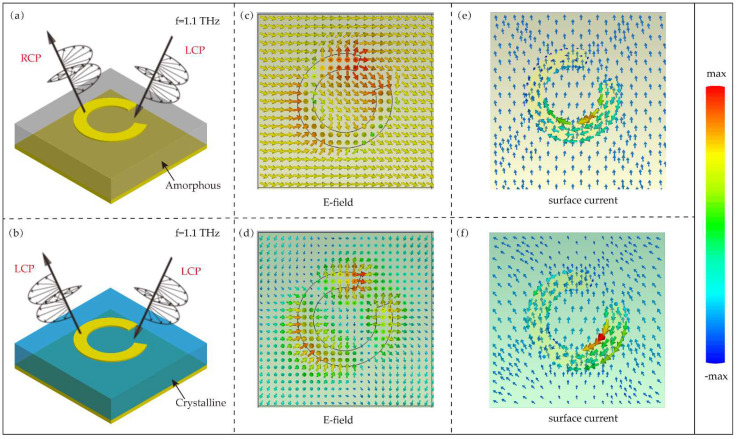
(**a**,**b**) The schematic diagram of opposite electromagnetic responses of unit cell in amorphous and crystalline states. (**c**,**d**) The top view of the electric field Ex distributions under the illumination of LCP in amorphous and crystalline states. The arrows represent the direction of the electric field. (**e**,**f**) The top view of the surface currents distributions under the illumination of LCP in amorphous and crystalline state. The arrows represent the direction of the surface current.

**Figure 6 nanomaterials-12-03592-f006:**
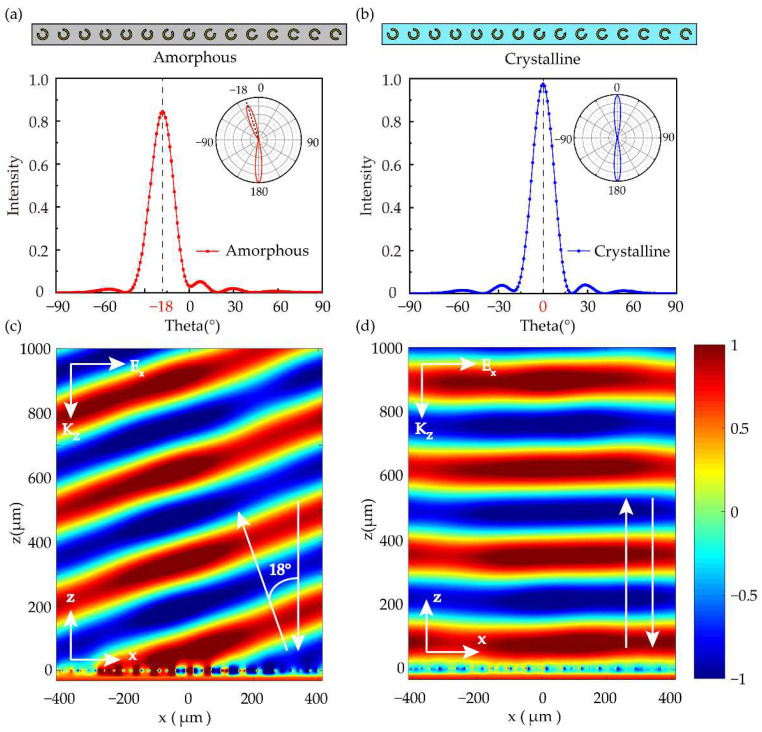
Simulated cross-polarization and co-polarization reflection intensity as a function of orientation angle *φ* in amorphous states (**a**) and crystalline states (**b**), where the insets are the simulated polar. The distributions of electric field *Ex* in the periodic unit cell under the illumination of LCP at the resonant wavelength of 1.1 THz in amorphous (**c**) and crystalline (**d**) states (the blue and red band represent the slight variation of phase).

**Figure 7 nanomaterials-12-03592-f007:**
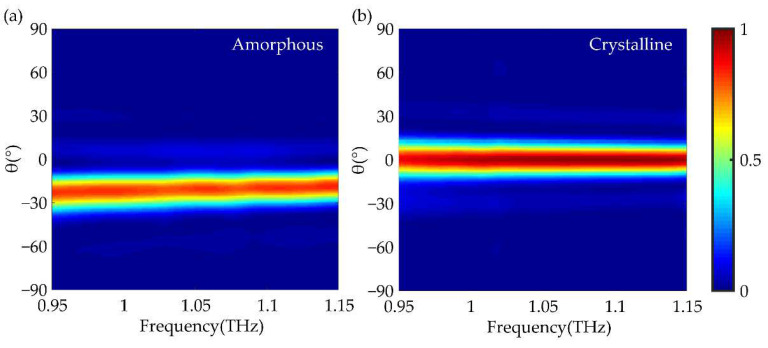
Normalized far-field radiation patterns of anomalous beam deflector when incidence light frequencies in 0.95~1.15 THz. The deflection angle and reflection intensity of the device in GST amorphous state (**a**) and crystalline state (**b**).

**Figure 8 nanomaterials-12-03592-f008:**
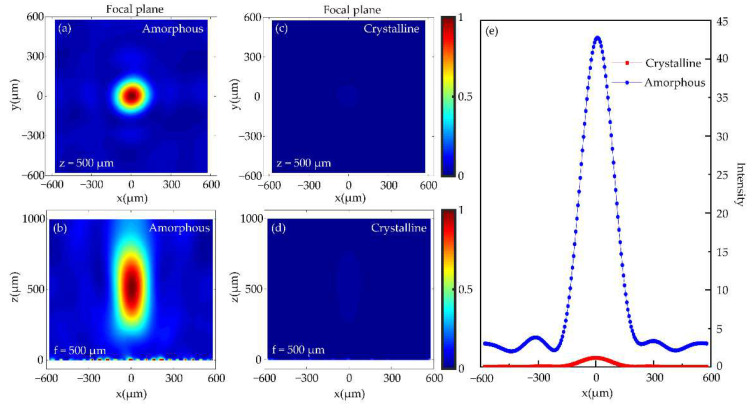
Under illumination of 1.05 THz LCP light, the sectional views of GST in amorphous (**a**,**b**) and crystalline (**c**,**d**) on focal plane z = 500 μm and plane y = 0 μm of the focusing metalens. (**e**) Electric field intensity distribution of z = 500 μm sectional view along *x*-axis in GST two states.

**Figure 9 nanomaterials-12-03592-f009:**
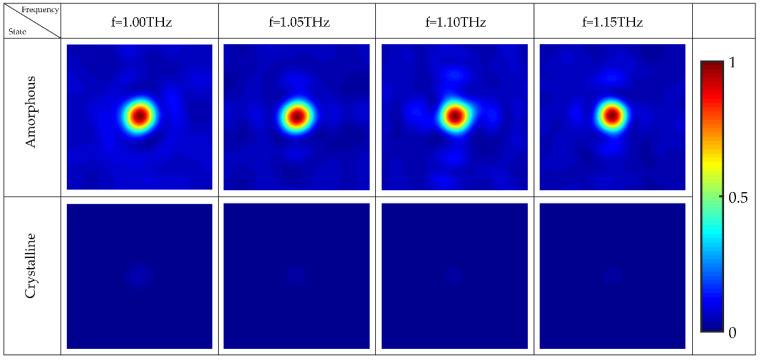
The focus situation of four reflective focusing metalenses at four frequencies in different GST states.

## Data Availability

The data underlying the results presented in this paper are not publicly available at this time but may be obtained from the authors upon reasonable request.

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
