# Peer review of "Tunable Terahertz Wavefront Modulation Based on Phase Change Materials Embedded in Metasurface"

_nanomaterials, 2022, doi:10.3390/nano12203592_

Round 1

Reviewer 1 Report (New Reviewer)

Here Zhang et al. present a tunable THz wavefront and describe the procedure obtaining this metamaterial surfaces embedded to the phase change material GST. This topic is potentially interesting and deserves attention. There are a lot of work and progress have been done as also the authors presented in their introduction.  This being said, I could not see why the authors approach should be more advantageous compared to the existing materials and how this work is providing important information for the field. Authors can also spend some time to explain certain things a bit more. I support the publication, but the following points should be addressed before. 

1. Are amorphous and crystalline states fixed from the begin with? If not, how the phase of GST is changed? If so, how this system can be tunable?

2. Authors use abbreviations unnecessarily, for instance, GA. I think it can be just used as it is in the long form, as it has not been used excessively and reduce the number of abbreviations. Easy to follow.

3. I don’t understand the advantage of using GST. Authors listed several advantage of GST over, for instance, VO2. However, the tunable frequency range of VO2 based materials are much larger than GST. For instance, Journal of Infrared, Millimeter, and Terahertz Waves volume 41607–631 (2020) gives an overview of these. Authors should explain more why their material is better.

Author Response

Response to the comments of reviewer 1:

Here Zhang et al. present a tunable THz wavefront and describe the procedure obtaining this metamaterial surfaces embedded to the phase change material GST. This topic is potentially interesting and deserves attention. There are a lot of work and progress have been done as also the authors presented in their introduction.  This being said, I could not see why the authors approach should be more advantageous compared to the existing materials and how this work is providing important information for the field. Authors can also spend some time to explain certain things a bit more. I support the publication, but the following points should be addressed before. 

  1. Are amorphous and crystalline states fixed from the begin with? If not, how the phase of GST is changed? If so, how this system can be tunable?

Response: GST can be deposited on the metallic ground by magnetron sputter, and the initial phase is amorphous. Besides, the tunability of our devices is technically accessible as the tunability of GST-based devices has been demonstrated by several groups. The mutual conversion of GST between amorphous and crystalline states is achieved by thermal energy or direct Joule heating. The commonly used conversion pathways include heat, light, and electricity. Taking Ge2Sb2Te5 as an example, it can be transformed into a crystalline state by heating it above 150°C. Heating crystalline Ge2Sb2Te5 to 640°C followed by rapid annealing can transform it from crystalline to amorphous. The transition time between the two states is extremely short, in the nanosecond range. With the excitation of electric pulses or laser pulses, the time required for the crystallization and amorphization processes is shorter, and even rapid transformation at the femtosecond level can be achieved. In addition, whether GST is in an amorphous state or a crystalline state, the material properties at room temperature are stable. Therefore, it is often used in fast and reconfigurable devices with high thermal stability, fast switching speed, and repeatable reading and writing.

The main purpose of this work is realizing tunable wavefront manipulation via the combination of metasurfaces with GST. Due to the lack of setups in our lab, the reconfiguration of this device is not demonstrated. However, two possible methods are provided and illustrated in below to realize the phase transition properties of the devices. To realize the phase transition of metadevices, several techniques used in above works can be adopted. Two possible methods are illustrated in the Figure R1. Figure R1 (a) gives a method based on phase-change electronic memory. The metasurfaces are made on the TiN electrodes (for converting the electronic energy to thermal energy). By applying the specific electrical stimulus, the phase state can be changed between two states, commonly the continuous pulses excitation for transition from amorphous to crystalline and ultrashort single pulse excitation for transition from crystalline to amorphous. The second method is based on laser-direct writing, which is illustrated in the Figure R1 (b). The re-amorphization of GST layer can be achieved by a single pulse with high power intensity, and the crystallization of GST layer can be achieved by pulse sequence with lower power intensity [4-6]. Additionally, different from thermal annealing, the laser pulses do not damage the gold patch-antenna. Utilizing a scan femtosecond (fs) laser, the pulse duration can be controlled by adjusting the scan speed, and then the reconfiguration of metadevice can be achieved. Other possible methods, like the conductive-AFM, ion-beam etching and so on, can also be applied to our metadevices.

Figure R1. Schematics for realization of the reconfiguration of metadevices. (a) the method based on phase-change electronic memory. (b) the method based on laser-direct writing.

  1. Authors use abbreviations unnecessarily, for instance, GA. I think it can be just used as it is in the long form, as it has not been used excessively and reduce the number of abbreviations. Easy to follow.

Response: Thanks for your kindly advice. We have changed the abbreviation to full name. The revised parts are highlighted in red in the manuscript.

  1. I don’t understand the advantage of using GST. Authors listed several advantage of GST over, for instance, VO2. However, the tunable frequency range of VO2 based materials are much larger than GST. For instance, Journal of Infrared, Millimeter, and Terahertz Waves volume 41, 607–631 (2020) gives an overview of these. Authors should explain more why their material is better.

Response: GST alloys as the typical phase change materials, have been used for many years in optical disk storage, which benefits from its non-volatility. As mentioned in the manuscript, compared to Phase change material VO2 (Terahertz Metamaterial Modulator Based on Phase Change Material Vo2. Symmetry-Basel 2021, 13, 2230.), GST not only has greater advantages in stability, conversion rate and non-volatile but also exhibits high refractive index and contrast and lower absorption loss in THz region, which enables it to be integrated into THz devices to achieve tunable functions. Besides, GST alloys are ideal materials for switchable or reconfigurable devices, such as thermal emitters, Fresnel zone plates, and absorbers. VO2 is preferable to implement a dynamic modulation response in THz frequencies for its electrical and optical characteristics near room temperature. VO2 can achieve an ultrafast and brutal reversible metal–insulator transition from insulator to metal provoked by thermal, electrical, or optical stimuli. There is a dramatic variation in the electrical and optical performance during the phase transition (4–5 order of magnitude changes on the electrical conductivity σ-VO2), so VO2 is a promising candidate in tunable MM devices at THz frequencies to achieve excellent modulation characteristics. Therefore, VO2 is of crucial importance and can be employed in THz devices to tailor various performances. Indeed, a lot of research fields have been focused on VO2-based tunable devices to obtain adjustable and switchable functions, such as reconfigurable THz filters, polarization converters, tunable THz absorbers, and so on. In short, VO2 and GST materials have their own advantages, and different materials are selected for different situations. In our work, we inserted the GST material into MIM-type metasurface to achieve tunable wavefront manipulation with non-volatility and high efficiency in both states.

Reviewer 2 Report (New Reviewer)

Good article.

Question. The paper considers the normal incidence of the wave. How will the characteristics of the device change in an inclined incidence?

Author Response

Response to the comments of reviewer 2:

Good article.

Question. The paper considers the normal incidence of the wave. How will the characteristics of the device change in an inclined incidence?

Response: Thanks for the comment. The inclined incidence is indeed a concern that need to be considered, because the metadevices we designed are reflective. Although the incident light and the outgoing light can be isolated by using a semi-reflective mirror. However, in many applications, in order to measure the adjustable wavefront control, the incident light needs to obliquely impinges the metadevices so that the incident light and the outgoing light will not interfere. Therefore, we simulated the unit cell in amorphous and crystalline states with the incident angle of [0°, 10°, 20°, 30°]. Figure R1 depicts the corresponding simulation results. As shown in figure, when GST is in amorphous state, the cross-polarized reflectance decreases and the co-polarized reflectance increases with the increase of incident angle. But the changes are slightly, which does not affect the spectral response of the unit cell, including the polarization conversion contrast and bandwidth. When GST is in crystalline state, the cross-polarized reflectance and the co-polarized reflectance are extremely same with different incident angle in the frequency range of 0.8-1.15 THz. However, the bandwidth becomes narrow with the increase of incident angle. The bandwidth is still in the acceptable range at the incident angle of 30 degrees, and the incident angle of 30° can already ensure that the outgoing light will not be disturbed by the incident light in practical applications.

Figure R1. The cross-polarized reflectance and the co-polarized reflectance in amorphous and crystalline states with the incident angle of [0°, 10°, 20°, 30°].

Reviewer 3 Report (New Reviewer)

The authors have presented a simulation work on adopting a phase change material (GST) integrated metasurface for THz wavefront manipulation. The study is useful and timely. However, the authors should address the following clarifications - 

Comment 1: (a) In Page 2, lines 60 - 61: The authors state that - "In addition, the GST exhibits high refractive index and 60 lower absorption loss contrast between the amorphous and crystalline states in THz region.." The absorption loss contrast is also significant between the phases of GST, > 1 order? Please verify. 

Comment 2: In the introduction part, the authors should present the experimental works on PCM integrated THz beam steering works and highlight the differentiation with the current manuscript. Some references: 

(a) Lin, Q.-W., Wong, H., Huitema, L., Crunteanu, A., Coding Metasurfaces with Reconfiguration Capabilities Based on Optical Activation of Phase-Change Materials for Terahertz Beam Manipulations. Adv. Optical Mater. 2022, 10, 2101699. https://doi.org/10.1002/adom.202101699

(b) Shoujun Zhang, Xieyu Chen, Kuan Liu, Haiyang Li, Yuehong Xu, Xiaohan Jiang, Yihan Xu, Qingwei Wang, Tun Cao, and Zhen Tian, "Nonvolatile reconfigurable dynamic Janus metasurfaces in the terahertz regime," Photon. Res. 10, 1731-1743 (2022). 

Comment 3: Page 4, line 130 - 133. The authors could use the measured properties of GST in amoprhous and crystalline phase in THz region. The scaling of material properties across widely different spectrum i.e. from optical to THz is generally not a accurate. Hence, in Page 4, line 135-136 - "The imaginary part of the permittivity in both states is close to 0, reflecting excellent feature that 136 low absorption loss of GST, which is vital for metadevices with high efficiency". This statement is not accurate and is due to assumption that low loss property of GST at IR region, remains unchanged even at THz, which is actually not the case.  There are literatures available on GST properties at THz spectrum as listed below, that the  authors can use. 

 - P. Pitchappa et al., "Chalcogenide phase change material for active terahertz Photonics", Adv. Mater. 2019, 31, 1808157.

 - K. Makino et al., "Terahertz spectroscopic characterization of Ge2Sb2Te5 phase change materials for photonics applications", J. Mater. Chem. C, 2019,7, 8209-8215

Comment 4: Page 4, line 160 - "After optimization, the polarization conversion efficiency and the working bandwidth increased significantly.", can the authors present the data to show the improvement through GA optimization in the appendix? and also state how much was the improvement. 

Comment 5: Page 4, line 186, the optimized thickness of GST is 20 um, but realistically cannot be fabricated. Can the authors provide insights into how such as device can be realized?

Comment 6: Page 7, Line 241 - In the amorphous case, the surface currents are antiparallel and hence should be a magnetic resonance and in the crystalline case, the surface currents on either side of SRR are parallel. But I am not sure, what is a symmetric and asymmetric mode and why are they required for polarization conversion. The authors should explain this point and provide relevant literature. 

Author Response

Response to the comments of reviewer 3:

The authors have presented a simulation work on adopting a phase change material (GST) integrated metasurface for THz wavefront manipulation. The study is useful and timely. However, the authors should address the following clarifications – 

Comment 1: (a) In Page 2, lines 60 - 61: The authors state that - "In addition, the GST exhibits high refractive index and lower absorption loss contrast between the amorphous and crystalline states in THz region…" The absorption loss contrast is also significant between the phases of GST, > 1 order? Please verify. 

Response: Thanks for your comment. In our simulation setup, the permittivity of GST is shown in Figure R1(a). In the broadband region, we can see that the real part of the GST permittivity exhibits a significant difference between amorphous and crystalline states. Figure R1(b) highlights the imaginary part of the permittivity, showing that the imaginary part of the permittivity is small but not 0 in the range of 0.1 ~ 2 THz in the two states, that is, our simulation results consider the case of absorption loss. Therefore, we performed the simulations in the lossy case. As shown in Figure R1(c)(d), the simulation results show high cross-polarization conversion and low co-polarization conversion in the amorphous state. However, it has low cross-polarization conversion efficiency and high co-polarization conversion efficiency in the crystalline state.

Figure R1. (a) The real and imaginary parts of permittivity of GST in two states, A and C represent the amorphous and crystalline states, respectively. (b) The imaginary part of permittivity. The simulated results of cross-polarization and co-polarization reflectance in amorphous state (c) and crystalline state (d).

Comment 2: In the introduction part, the authors should present the experimental works on PCM integrated THz beam steering works and highlight the differentiation with the current manuscript. Some references: 

(a) Lin, Q.-W., Wong, H., Huitema, L., Crunteanu, A., Coding Metasurfaces with Reconfiguration Capabilities Based on Optical Activation of Phase-Change Materials for Terahertz Beam Manipulations. Adv. Optical Mater. 2022, 10, 2101699. https://doi.org/10.1002/adom.202101699

(b) Shoujun Zhang, Xieyu Chen, Kuan Liu, Haiyang Li, Yuehong Xu, Xiaohan Jiang, Yihan Xu, Qingwei Wang, Tun Cao, and Zhen Tian, "Nonvolatile reconfigurable dynamic Janus metasurfaces in the terahertz regime," Photon. Res. 10, 1731-1743 (2022). 

Response: Metasurfaces for PCM integration in terahertz switchable metalens, optical vortex generators, and beam steering have been extensively studied, and there are also some experimental works to verify these properties. Here, our design combined genetic algorithm used PB phase and GST, and shows high contrast of polarization conversion in two states. Besides, the wavefront manipulation with PB phase can accurately control the abrupt phase by rotating orientation angle. Therefore, our design is sophisticated, and the optimization strategy can relief the difficulty of design, which is the novelty of our work. Perhaps the introduction of experimental work and difference is less in our manuscript, we add the following sentence into the second paragraph of introduction: “Among them, PCMs are integrated in THz switchable metalens, optical vortex generators and beam steering verified tunable wavefront manipulation [34, 35], and experimental verifications were carried out.”

In addition, we add the following sentence into the third paragraph of introduction: “The design in our work is technically challenging as it requires high contrast of polarization conversion between two states. We utilized the genetic algorithm (GA) to optimize the geometric parameters of the unit cell and to obtain the high contrast response in two states.”

Comment 3: Page 4, line 130 - 133. The authors could use the measured properties of GST in amoprhous and crystalline phase in THz region. The scaling of material properties across widely different spectrum i.e. from optical to THz is generally not an accurate. Hence, in Page 4, line 135-136 - "The imaginary part of the permittivity in both states is close to 0, reflecting excellent feature that 136 low absorption loss of GST, which is vital for metadevices with high efficiency". This statement is not accurate and is due to assumption that low loss property of GST at IR region, remains unchanged even at THz, which is actually not the case.  There are literatures available on GST properties at THz spectrum as listed below, that the authors can use. 

 - P. Pitchappa et al., "Chalcogenide phase change material for active terahertz Photonics", Adv. Mater. 2019, 31, 1808157.

 - K. Makino et al., "Terahertz spectroscopic characterization of Ge2Sb2Te5 phase change materials for photonics applications", J. Mater. Chem. C, 2019,7, 8209-8215

Response: Thank you for pointing out this. The measured permittivity of GST is the fitting results by spectroscopic ellipsometer (SE) via a mathematic approach, which may change with different models or starting value assignments. The characterization of metadevices in crystalline state (i.e. the suppression of cross-polarized reflectance) is determined by the real part of permittivity of GST layer, and the influence caused by imaginary part is slight. Besides, the permittivity of GST shows a high contrast between the real and imaginary part. However, to rigorously characterize the permittivity in simulations, we re-fitted the permittivity by SE and the results is closer to the reference [P. Pitchappa et al., "Chalcogenide phase change material for active terahertz Photonics", Adv. Mater. 2019, 31, 1808157.], [ K. Makino et al., "Terahertz spectroscopic characterization of Ge2Sb2Te5 phase change materials for photonics applications", J. Mater. Chem. C, 2019,7, 8209-8215]. We have replotted the Figure R2, which exhibits better agreement between simulated results and measured results in crystalline state. In previous simulations, the deviation between simulated results and measured results caused by extinction coefficient of GST was also discussed in the manuscript, which was verified by the new simulated results. Thank you for pointing out this to improve the manuscript.

In previous, we have measured a lot of data on the permittivity of GST, as shown in Figure R2(a-d), including visible light from 380 to 780 nm and infrared light from the range of 0.78 ~ 16 um in the crystalline and amorphous states, respectively. Not only the measurement range is very wide, but also there is a sufficient amount of measurement data. According to these large amounts of discrete data, these data are input into the CST software for fitting, and this fitting is applied in the crystalline and non-crystalline state. The results in the crystalline state are very close to the references mentioned by reviewer. The fabrication method for GST layer in our previous work is different from that of reference. Therefore, there will be a slight difference in the permittivity of GST, but which is within an acceptable range. Additionally, we have added the following sentence to the manuscript: “The permittivity is fitted to the range of 0.1 ~ 2 THz from the measured data of visible light and infrared by CST software utilizing enough measurements, and the result is consistent with the previous papers [37, 38].”

Figure R2. Measured permittivity of GST film. (a) and (b) the real and imaginary part for both amorphous and crystalline states in the wavelength range 0.3 to 2.5µm measured by SENTECH SE850 (c) and (d) in wavelength range 2.5 to 16µm measured by SENTECH SENDIRA. (e) the real and imaginary parts of the permittivity in the range of 0.1 ~ 2 THz in both amorphous and crystalline states. (f) the imaginary part of the permittivity in both amorphous and crystalline states.

Comment 4: Page 4, line 160 - "After optimization, the polarization conversion efficiency and the working bandwidth increased significantly.", can the authors present the data to show the improvement through GA optimization in the appendix? and also state how much was the improvement. 

Response: The optimization strategy combined genetic algorithm can help us quickly realize the design of metasurface, instead of relying on the designer's experience and a lot of time and energy consumption. As you suggested, the previous description may not be accurate enough to explain the optimization process. For ease of understanding, we modify the sentence as follows: “After the rough model of the metasurface unit cell is determined, a set of optimal parameters need also to be simulated to determine the final unit structure. In order to avoid a lot of time and energy consumption by manual parameter setting, we use genetic algorithm to automatically optimize a set of optimal parameters, and just need to focus on the final result. Meanwhile, the optimal parameters represent the final unit structure showing high polarization conversion efficiency and wide working band-width.”

Comment 5: Page 4, line 186, the optimized thickness of GST is 20 um, but realistically cannot be fabricated. Can the authors provide insights into how such as device can be realized?

Response: The thickness 20 um of GST can be deposited on metallic ground by magnetron sputter in theory. In our previous work, we designed three metadevices with switchable photonics spin-orbit interactions, whose structure includes GST layer with thickness of 600nm. To obtain the GST layer, we took about 25 minutes to fabricate 600 nm GST by utilizing magnetron sputter, so the sputtering speed of GST is about 24 nm/min. Theoretically, 20um-thick GST takes 13.9 hours to obtain by magnetron sputtering. Due to the lack of setups in our lab, the fabrication of this device is not demonstrated. Therefore, this manuscript focuses on the realization of the THz wavefront modulation function and new construction.

Comment 6: Page 7, Line 241 - In the amorphous case, the surface currents are antiparallel and hence should be a magnetic resonance and in the crystalline case, the surface currents on either side of SRR are parallel. But I am not sure, what is a symmetric and asymmetric mode and why are they required for polarization conversion. The authors should explain this point and provide relevant literature. 

Response: Thanks for the comment. In the manuscript, we depicted cross-polarized and co-polarized efficiency in amorphous and crystalline states to show the distinct electromagnetic responses. Besides, the polarization conversion ratio (PCR) and phase difference (PD) also verify the high contrast of polarization conversion efficiency in amorphous and crystalline states, which proves that the designed metasurface with GST possesses the capability of tunable wavefront manipulation. In order to further reveal the physical mechanism of the unit cell producing opposite electromagnetic responses in GST two states, we illustrate the instantaneous electric field distributions and surface current under normal incidence. Figure R3a, b illustrated the opposite electromagnetic responses of the two devices under different states. In amorphous state, the designed unit cell can realize the circular polarization conversion. In crystalline state, the unit cell works as a mirror-like device. Figure R3c, d show the distributions of electric field Ex at the resonant frequency of 1.1 THz in the xy plane, respectively. The direction of arrow represents the vibration direction of electric field vector. It is obvious that the electric fields are highly localized in the opening of the split ring in both states. The vibration direction of electric field vector rotates at the opening of the split ring in amorphous state, but in crystalline state, the electric field directions on both sides of the opening of the split ring are opposite. The different electric field distributions lead to diametrically opposite polarization conversion efficiencies in the two states. Figure R3e, f show the distributions of surface current at the resonant frequency of 1.1 THz in the xy plane, and the direction of arrow represents the direction of surface current. In amorphous state, the surface currents have the same circulating direction along the split ring, corresponding to the symmetric resonance mode. Since the direction of the surface current in the amorphous state is the same as that on the split-ring resonator, the incident circularly polarized light will be deflected along the direction of the surface current to realize polarization conversion. But in crystalline state, the surface currents exhibit both counter-clockwise and clockwise directions along the split ring, corresponding to the asymmetric resonance mode.

The asymmetric resonance mode induces the magnetic resonance. Therefore, the components of the incident circularly polarized light deflecting along the left and right sides of the split ring resonator will repeal by implication, and THz waves are simply specular reflected. The much larger refractive index of GST suppresses the transversal coupling. Thus, the unit cell exhibits a weak anisotropy and low polarization conversion efficiency. Based on this characteristic, the tunable wavefront manipulation can be realized by appropriately arranging the unit cell. The analysis about the surface current that causes polarization conversion or simply specular reflection refers to the reference [Resonance coupling and polarization conversion in terahertz metasurfaces with twisted split-ring resonator pairs[J]. Optics Express, 2017, 25(21):25842, and Terahertz broadband polarization converter based on the double-split ring resonator metasurface[J]. SN Applied Sciences, 2021, 3(9):1-7.]. The corresponding parts in manuscript are revised and highlighted in red, and the reference are also mentioned in the manuscript.

Figure R3. (a) and (b) the schematic diagram of opposite electromagnetic responses of unit cell in amorphous and crystalline states. (c) and (d) The top view of the electric field Ex distributions under the illumination of LCP in amorphous and crystalline states. (e) and (f) The top view of the surface currents distributions under the illumination of LCP in amorphous and crystalline state.

Round 2

Reviewer 3 Report (New Reviewer)

The authors have addressed the comments. 

This manuscript is a resubmission of an earlier submission. The following is a list of the peer review reports and author responses from that submission.

Round 1

Reviewer 1 Report

The paper entitled Tunable terahertz wavefront modulation based on phase change materials embedded in plasmonic metasurfaces proposes to use a combination of a plasmonic metasurface and phase change material (PCM) in order to allow binary manipulation of the wavefront in the THz regime. The authors use a metal-insulator-metal metasurface with a c-shape split-ring resonator (CSRR); this allows them to obtain high conversion efficiency of circular polarization to the cross-polarized state in the amorphous state, while this conversion efficiency is very low when the PCM is in the crystalline phase. The phase shift of the effective medium is shown to be close to π on a large spectrum in the amorphous state, which is required for the efficient conversion of circular polarization state to its cross-polarized state, while the phase shift is close to 0 in the crystalline state. Using the Pancharatnam-Berry phase (PB), the authors successfully simulate beam steering with a linear gradient of orientation of the split-ring resonator on a large bandwidth as well as beam focusing by a suitable distribution of the orientation of the split-ring resonator in each unit cell; these effects can be turned on or off by switching the phase change material from the crystalline to amorphous phase.

The proposed ideas of combining metasurfaces with a PCM and using the PB phase to control the wavefront had previously been proposed by Zhou et al. J. Phys. D: Appl. Phys. 53, 204001 (2020), using rod-shaped instead of CSSR. Here, the only difference with Zhou lies in the use of CSSR. Once it is established that the unit cell behaves optically as a halfwave plate in one phase and as an isotropic material in the other phase of the PCM, then all the results of the simulations are straightforward and can be deduced from other studies.

Besides, split-ring resonators have been proposed and used extensively in metamaterials to produce a magnetic resonance, but here the C-shape is merely used to produce optical anisotropy; the different shapes of metal units that can produce such anisotropy is countless. Moreover, it is not clear where the plasmonic response come into play: the proposed material, copper, does not support surface plasmon polariton, and the plasmonic resonance should come at a cost of absorption. The information conveyed by Fig. 4, with the electric field distribution is not clear: how does it help the reader or the designer to figure out the effective response of the material for each of the two phases?

The authors should focus their attention on what makes the originality of their work. Why did they choose a split-ring resonator? What is their methodology to adjust the parameters to obtain a flat response with frequency and π and 2π phase shift in the two phases respectively? Once the latter is established, all the results in section 3 go without saying and should not be the main topic of the paper. How does the plasmonic response enter in the mechanism of the optical response? Finally, only simulation results are presented in this paper. Do what extent do the authors expect the actual device to perform as the simulation results predict?

End of the report.

Reviewer 2 Report

The paper is interesting and worthwhile for publication.

The idea to control the retardation of the phase and of the intensity controlling the azimuthal position of the partially complete rings.

The authors have clearly described also the dependence of the response on the substrate and also on the angular dependence.

 I have carefully review the paper that in my opinion is free from mistakes and I again suggest the publication.

Reviewer 3 Report

This is an interesting paper.  The search for tunable wavefront manipulation is of high interest, and the modeling has been carried out thoroughly.  My only suggestion is that it would be highly desirable to show some experimental verification of these predictions.

Reviewer 4 Report

The authors report on a numerical design study for THz-metasurfaces operating with GST to provide tunability after finalizing the design. While the overall idea is indeed very convincing the paper lacks novelty as basically the same idea with almost the same design has recently been published: Zhang et al. PhotoniX (2022) 3:7, https://doi.org/10.1186/s43074-022-00053-5

In this paper the authors not only demonstrate numerical simulations but also experiments on fabricated samples, showing that the method indeed works. As the differences are found in minor details (gold instead of aluminum), I do not see enough scientific novelty to warrant the publication of this manuscript in its current form. 

In general, the English language needs a lot of editing. I suggest using the tool "grammarly". Furthermore, Figure numbers are not always corresponding to the correct figures. This has to be fixed. 

However, in light of the above-mentioned publication, I do not see this manuscript as fit for publication.